# Germline determinants of humoral immune response to HPV-16 protect against oropharyngeal cancer

Aida Ferreiro-Iglesias [1✉], James D. McKay[2], Nicole Brenner [3], Shama Virani [1], Corina Lesseur [1,4], Valerie Gaborieau[1], Andy R. Ness[5,6], Rayjean J. Hung [7], Geoffrey Liu[8], Brenda Diergaarde[9,10], Andrew F. Olshan[11], Neil Hayes [12], Mark C. Weissler[13], Lea Schroeder[3], Noemi Bender[3], Michael Pawlita [3], Steve Thomas[6], Miranda Pring[6], Tom Dudding [6], Beatriz Kanterewicz[10], Robert Ferris[10], Sera Thomas[7], Yonathan Brhane[7], Virginia Díez-Obrero[14], Maja Milojevic [1], Karl Smith-Byrne [1], Daniela Mariosa [1], Mattias J. Johansson [1], Rolando Herrero[15], Stefania Boccia[16,17], Gabriella Cadoni[18,19], Martin Lacko [20], Ivana Holcátová[21], Wolfgang Ahrens[22], Pagona Lagiou[23], Areti Lagiou[24], Jerry Polesel [25], Lorenzo Simonato[26], Franco Merletti[27], Claire M. Healy[28], Bo T. Hansen[29], Mari Nygård[30], David I. Conway[31], Sylvia Wright[32], Tatiana V. Macfarlane[33], Max Robinson [34], Laia Alemany[35,36], Antonio Agudo[35], Ariana Znaor [37], Christopher I. Amos[38], Tim Waterboer[3,39] & Paul Brennan [1,39✉]

Although several oropharyngeal cancer (OPC) susceptibility loci have been identified, most previous studies lacked detailed information on human papillomavirus (HPV) status. We conduct a genome-wide analysis by HPV16 serology status in 4,002 oral cancer cases (OPC and oral cavity cancer (OCC)) and 5,256 controls. We detect four susceptibility loci pointing to a distinct genetic predisposition by HPV status. Our most notable finding in the HLA region, that is now confirmed to be specific of HPV(+)OPC risk, reveal two independent loci with strong protective effects, one refining the previously reported HLA class II haplotype association. Antibody levels against HPV16 viral proteins strongly implicate the protective HLA variants as major determinants of humoral response against L1 capsid protein or E6 oncoprotein suggesting a natural immune response against HPV(+)OPC promoted by HLA variants. This indicates that therapeutic vaccines that target E6 and attenuate viral response after established HPV infections might protect against HPV(+)OPC.

A full list of author affiliations appears at the end of the paper.

Oropharyngeal cancers (OPC) are a heterogeneous group of tumors with some caused by smoking and excessive alcohol consumption and others by human papillomavirus (HPV) infections. The majority of HPV-positive OPC are caused by HPV type 16 (HPV16), with HPV surpassing tobacco exposure as the leading cause of OPC in many countries[1,2]. HPV16-positive (HPV(+)) OPC are now considered a separate disease entity, with different treatment regimen, and different biological, epidemiological, clinical, and prognostic characteristics from HPV16-negative (HPV(−)) OPC[1,3–6].

Germline genetic variants are likely to play different roles in susceptibility to HPV(+) and HPV(−) OPC and identification of specific variants may help in the development of effective prevention, screening, and treatment strategies. Genome-wide association studies (GWAS) have identified several susceptibility loci for OPC, but have generally been limited by lack of information on HPV status[7–9]. In our previous study, the effect of the human leukocyte antigen (HLA) region (6p21.3), specifically the HLA haplotype *DRB1\*13:01-DQA1\*01:03-DQB1\*06:03*, was found to be protective for OPC[9]. The same haplotype was previously reported to be protective for cervical cancer[10], and this further highlights the need for comprehensive genetic analyses of OPC by HPV status.

To achieve this, correct identification of HPV-driven oropharyngeal tumors is critical. HPV serology is an appropriate solution in situations where tumor tissue for direct assessment is not available. It captures humoral immune response levels via measurement of antibodies against viral proteins. Antibodies against HPV L1 are considered to represent cumulative past and present HPV infection from multiple possible anatomic sites (i.e., genital, anal, or oral), and are relatively common in individuals without an HPV-related tumor[2,6,11–13]. Antibodies against HPV E6 and E7 oncoproteins occur in response to an underlying HPV-driven neoplastic process and are therefore expected to be at high levels among those with an HPV-driven tumor and low levels among cancer-free individuals. Large population-based cohort studies have demonstrated the potential of serology measures against HPV16 early viral proteins to detect individuals with HPV16-driven OPC with a high sensitivity and specificity[14,15]. In particular, HPV16 E6 antibodies have been shown to occur in more than 90% of those with HPV16(+) OPC while seroprevalence among population controls is a rare condition (<1%)[2,14,16,17]. Consequently, antibodies against specific E proteins are useful in determining tumor HPV status[2,6,16]. In addition, they may also provide information on what humoral responses are activated in response to infection and malignant progression.

In this work, we conduct a GWAS in OPC stratified by HPV status based on HPV16 serology to examine genetic architecture, predisposition, and potential functional consequences of the associated loci to develop insights into OPC etiology. Our study identifies OPC genetic heterogeneity by HPV16 status and implicates the HPV(+)OPC protective HLA variants as major determinants of humoral response against viral proteins.

## Results

Following stringent quality-control steps on genotyping data[9], 1643 OPC cases and 5256 cancer-free controls remained for analyses (Table 1, Supplementary Tables 1–3). The OC case series comprised 2359 cases and was used to complement HPV(−)OPC analyses given their common risk factors (Supplementary Table 4).

Based on the serology data, 65.6% of the OPC cases (1078 of 1643) were classified as HPV(+). Among the controls with serology data available (1543), 16 (1%) were classified as seropositive

**Table 1 Background characteristics of the participating subjects after quality control.**

| | Oropharyngeal cancer cases no. (%) | Control no. (%) | p value[a] |
|---|---|---|---|
| **Oncoarray individuals passed QC** | **1643** | **5256** | |
| Geographic region | | | 0.004 |
| Europe | 730 (44.4) | 2927 (55.7) | |
| North America | 913 (55.6) | 2329 (44.3) | |
| Age | | | 0.07 |
| ≤50 | 293 (17.84) | 889 (16.91) | |
| 51–60 | 687 (41.84) | 1561 (29.7) | |
| 61–70 | 486 (29.6) | 1649 (31.37) | |
| >70 | 176 (10.72) | 1157 (22.01) | |
| Unknown | 1 | 0 | |
| Sex | | | <0.001 |
| Male | 1332 (81.1) | 3334 (63.4) | |
| Female | 311 (18.9) | 1922 (36.6) | |
| Unknown | 0 | 0 | |
| Smoking status | | | <0.001 |
| Never | 391 (25.9) | 2019 (40.9) | |
| Former | 640 (42.4) | 1821 (36.9) | |
| Current | 479 (31.3) | 1096 (22.2) | |
| Unknown | 133 | 320 | |
| Drinking status | | | 0.09 |
| Never | 249 (16.8) | 846 (17.8) | |
| Ever | 1233 (83.2) | 3898 (82.2) | |
| Unknown | 161 | 512 | |
| HPV16 status | | | |
| HPV16 seropattern | | | <0.001 |
| Positive | 1078 (65.6) | 16 (1.0) | |
| Negative | 565 (34.4) | 1527 (90.0) | |
| Unknown | 0 | 3713 | |
| HPV 16 E6 (cutoff:1000 MFI) | | | <0.001 |
| Positive | 1059 (64.5) | 15 (1.0) | |
| Negative | 584 (35.5) | 1528 (99.0) | |
| Unknown | 0 | 3713 | |
| HPV 16 L1 | | | <0.001 |
| Positive | 767 (46.7) | 64 (4.2) | |
| Negative | 875 (53.3) | 1477 (95.8) | |
| Unknown | 1 | 3715 | |

OPC cases definition: C01.9, C02.4, C05.1, C05.2, C09.0-C10.9.
[a]Logistic regression models were implemented to test associations of each variable

for HPV16; thus, the population specificity was 99.0% (95% CI: 98.3–99.4%). Median fluorescence intensity (MFI) values of HPV16 serology markers (E1, E2, E7) were markedly higher in cases compared to controls (Supplementary Table 5).

GWA analyses stratified by HPV16 status (1078 HPV(+) and 565 HPV(−) OPC cases, and 5256 controls) within each continent and meta-analyzed yielded 7,574,753 SNPS for risk association analyses. GWAs results of the two traits are presented in Fig. 1. There was no evidence of genomic inflation for any of the analyses after scaling it to 1000 cases and controls[9] (Supplementary Fig. 1). All findings with a P value less than $1 \times 10^{-6}$ are reported in Supplementary Data 1–3.

**The HLA region is significantly associated with HPV(+)OPC risk.** A strong signal ($P = 4.5 \times 10^{-11}$) mapped to 6p21.32 was identified for HPV(+)OPC cases (Fig. 1; Table 2). This area of the genome encodes the HLA region, which regulates the immune response. In addition, two suggestive associations at locus 3q26.1

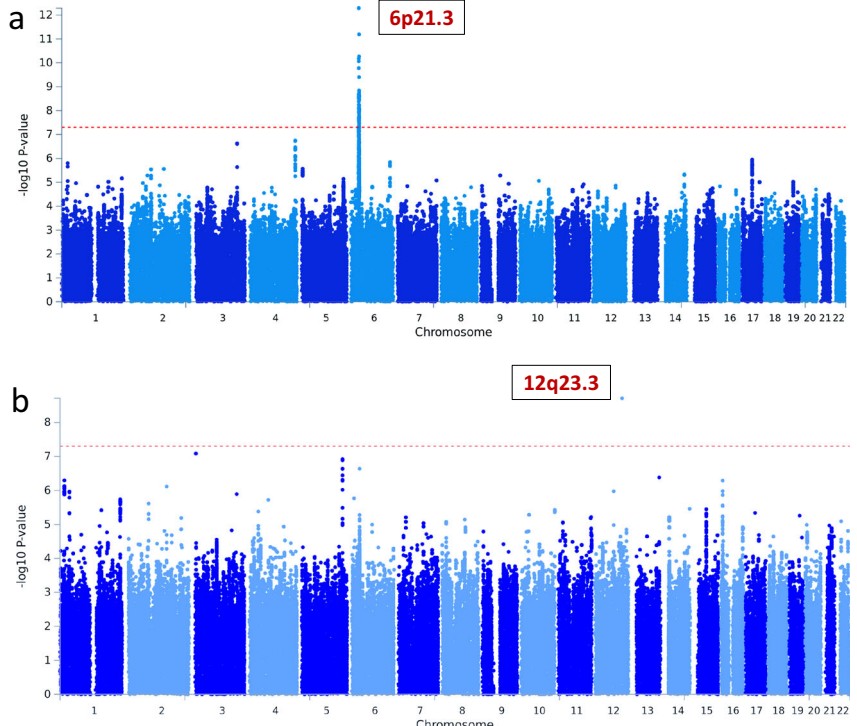

**Fig. 1 Genome-wide association regional meta-analysis results.** Red line correspond to $P = 5 \times 10^{-8}$. The y axes show $-\log 10$ P values. **a** HPV(+)OPC cases analysis with 1078 cases and 5256 controls. **b** HPV(−)OPC cases analysis with 565 cases and 5256 controls. GWAs for each world region were performed using multivariable unconditional logistic regression assuming a log-additive genetic or dosage model with age, sex, and eigenvectors as covariates. P-values are shown from fixed-effect meta-analysis of regional association statistics. HPV human papillomavirus, OPC oropharyngeal cancer.

**Table 2 Top genome-wide significant regions and their most likely functional variants from regional meta-analyses of oropharyngeal cancers by HPV16 seropattern status.**

| Independent regions | Locus | Variant | Info[a] | EA/OA[b] | Frequency (%) Controls/Cases | Meta-analysis[c] OR (95% CI) | P value | Q_p |
|---|---|---|---|---|---|---|---|---|
| HPV(+)OPC[d] HLA class I | | | | | | | | |
| | Intergenic | rs4713462 | genotyped | A/G | 32.6/21.3 | 0.66 (0.59–0.75) | $4.5 \times 10^{-11}$ | 0.19 |
| | HLA-B | *1501* | 0.97 | P/A | 5.9/3.0 | 0.44 (0.33–0.58) | $1.13 \times 10^{-8}$ | 0.20 |
| | | 156 -Trp | 0.98 | P/A | 6.1/3.2 | 0.45 (0.34–0.59) | $1.05 \times 10^{-8}$ | 0.21 |
| HLA class II | Class II haplotype | DRB1*1301-DQA1*0103-DQB1*0603 | | P/A | 6.7/2.6 | 0.42 (0.31–0.56) | $7.4 \times 10^{-9}$ | 0.86 |
| | HLA-DRB1 | *1301* | 0.99 | P/A | 6.7/2.8 | 0.43 (0.32–0.58) | $1.44 \times 10^{-8}$ | 0.95 |
| | | 71 -Glu[e] | 0.94 | P/A | 14.5/7.9 | 0.56 (0.47–0.68) | $2.8 \times 10^{-9}$ | 0.30 |
| | HLA-DQA1 | *0103* | 1 | P/A | 7.8/3.1 | 0.45 (0.34–0.59) | $8.7 \times 10^{-9}$ | 0.90 |
| | HLA-DQB1 | *0603* | 0.98 | P/A | 7.3/3.1 | 0.45 (0.34–0.60) | $3.1 \times 10^{-8}$ | 0.35 |
| HPV(-)OPC[f] 12q23.3 | BTBD11 | rs35189640 | 0.94 | T/C | 2.0/4.7 | 2.73 (1.97–3.79) | $1.1 \times 10^{-9}$ | 1 |

*HLA* human leukocyte antigen, *OR* odds ratio, *95% CI* confidence interval, *Q_p* Cochran's heterogeneity p value
[a]Imputation INFO (R2) is the average across imputation batches.
[b]Effect allele/other allele; P stands for presence and A stands for absence for amino acid polymorphisms and HLA alleles.
[c]Regional meta-analyses Europe and North America.
[d]Number of subjects: 1078 HPV(+)OPC and 5256 controls.
[e]amino acid change due to rs9269942 C/A [Ala (GCG) --> Glu (GAG)].
[f]Number of subjects: 565 HPV(−)OPC negative and 5256 controls.

and 4q35.1 were detected just below the GWAS significance level ($P < 1 \times 10^{-7}$, Supplementary Table 6).

Stratified analyses, by study characteristics are summarized in Supplementary Fig. 2. No evidence of differences in effect size between characteristic strata were observed.

Of note, in this larger study population with HPV status information, we also replicated the previously observed association of three HLA-specific alleles (*DRB1*13:01*, *DQA1*01:03*, and *DQB1*06:03*) and their haplotype with decreased HPV(+)OPC risk (Table 2 and Supplementary Fig. 2)[9].

**Fine-mapping of the HLA region reveals two independent loci associated with risk of HPV(+)OPC.** The HLA region is highly polymorphic with extensive linkage disequilibrium (LD) and assessing the causality of identified associations is difficult. To fine-map the region, we therefore imputed sequence variation in classical HLA genes. The final set of imputed variants used in association analyses were of high quality; 92.5% of the variants had $R^2 \geq 0.9$, and 68.2% of the less common variants (MAF < 0.05) had $R^2 \geq 0.9$.

Associations with risk of HPV(+)OPC extended into the class I and II regions (Fig. 2a). The most significant variant was the A allele of rs4713462, an HLA class I intergenic variant that maps 5.6 kb 5′ of AL671883.1. This variant had not been identified in the previous GWAs analysis[9] in which HPV status data was available for only a limited number of cases. (OR = 0.66, $P = 4.5 \times 10^{-11}$; Fig. 2a; Table 2). Given its proximity with HLA-B and -C, we examined pairwise LD statistics across single nucleotide polymorphisms (SNPs) covering this region in our data. In Supplementary Fig. 12, pairwise D′ and $R^2$ indicate that SNP rs4713462 showed generally modest correlation with SNPs in HLA-B and C coding regions or surrounding areas. This is in line with our results in Supplementary Fig. 13, where rs4713462 is the most significant variant in the region with modest LD and association statistics with the variants covering HLA-C and B. Functional annotation revealed that rs4713462 is predicted to be an eQTL for *HLA-C* across different tissues including upper digestive tract tissues. Given the extended LD in the MHC region and the considerable number of variants below genome-wide level significance showing similar associations with *HLA-C* mRNA levels (Supplementary Table 7), we explored *HLA-C* expression genetic control across the MHC region. There did not appear to be a consistent association between the rs4713462 A allele and *HLA-C* gene expression in oral related tissues (Supplementary Fig. 3). This may suggest that HLA-C expression alone is unlikely to be mediating this association.

After conditioning on rs4713462, evidence remained of another protective association for the HLA class II region ($P < 5 \times 10^{-8}$). This mapped to an A allele of rs9269942, a multiple nucleotide variation that results in an amino acid change to glutamine in codon 71 (71-Glu) of *HLA-DRB1* protein (OR = 0.56, $P = 2.8 \times 10^{-9}$; Fig. 2b, Table 2) and translates to a location in the peptide-binding groove of *HLA-DRB1*13:01* and in three other *HLA-DRB1* alleles (*04:02, *13:02 and *11:02) for which a protective trend was also identified (Supplementary Fig. 4; Supplementary Table 8). This result further reinforces the possible functional role of this variant and refines the association of the HLA haplotype *DRB1*13:01-DQA1*01:03-DQB1*06:03* identified in our previous GWAs analysis for OPC[9] and with decreased cervical cancer risk[10].

When both rs4713462 and *HLA-DRB1* 71-Glu were conditioned on, no remaining signals were detected (Fig. 2c).

In addition to identifying loci associated with HPV-positive OPC risk, haplotype analysis was used to define the boundaries of associated variants within the HLA region given its complexity and extended LD. The only haplotype found associated with OPC risk was the HLA class II haplotype, *DRB1*13:01-DQA1*01:03-DQB1*06:03* (Table 2; Supplementary Fig. 1a). No other haplotype showed a significant association when considering Class I or both Class I and Class II blocks. Conditional analyses considering *DRB1*13:01-DQA1*01:03-DQB1*06:03* and rs4713462 in the same model, also revealed the independence of both effects (Table 3).

We subsequently assessed whether the independent variants identified to be protective of OPC were indicative of a protective effect from a haplotype, from specific amino acids within haplotypes, or from the variants themselves which could be

functionally mapped to other variants unrelated to HLA (Supplementary Table 11). To refine the associations within the MHC region, model selection was used to identify the combination of variant, amino acid and/or haplotype that best explained the data using BIC (Bayesian Information Criterion) as model selection criterion. Table 3 displays the combined models with the best fit to the data from this search. Model A was considered the best fitting model. This implies that the amino acid 71-Glu in HLA-DRB1*13:01 explains the association of the class II haplotype *DRB1*13:01-DQA1*01:03-DQB1*06:03* with risk of HPV(+)OPC (Table 2; Table 3; Supplementary Fig. 2a). Although Glu-71 accounted for the main part of the effect of DRB1-DQA1-DQB1 as it fitted the data in the conditional model and it has biological significance, amino acid (Ala-74) does not seem to be an independent effect on Glu-71 since the effect was not significant ($p > 5 \times 10^{-8}$) and weaker when both are in the model (Supplementary Table 12, model E), whereas the effect was stronger and significant in the absence of Glu-71 (Supplementary Table 12, model C) and in the individual model (Supplementary Table 11). For that reason, we cannot exclude additional loci within the haplotype supporting or contributing additively to risk, as probably is happening with the amino acid position 74 in HLA-DRB1*1301, also located in the protein binding groove (Supplementary Fig. 4).

**Association of each HLA variant with antibodies against viral proteins suggests differential HLA response to viral proteins.** We conducted a protein quantitative trait loci (pQTL) analysis of 8291 MHC imputed variants using HPV16 E6 or L1 antibody levels in all OPC patients to explore associations between these variants and antibody levels. Rs4713462 and 71-Glu were the variants most strongly associated with E6 and L1 antibody levels, respectively (Fig. 3, Supplementary Fig. 9). Specifically, results showed that the rs4713462 allele (A) was exclusively and strongly associated (B = −0.31, $P = 1.3 \times 10^{-8}$) with lower E6 antibody levels and not with L1 MFI levels (B = −0.07, $P = 0.05$); 71-Glu, which is located in the peptide-binding groove of *HLA-DRB1*13:01*, showed a robust association (B = −0.30, $P = 7.0 \times 10^{-8}$) with lower L1 MFI levels but no correlation with E6 MFI levels (B = −0.12, $P = 0.11$). This suggests that their association with HPV(+)OPC could potentially be mediated via influencing humoral immune response during viral tumorigenesis and corroborates the independence of these variants in their associations with lower risk of HPV(+)OPC.

We repeated the analyses for the top associated variants in the subset of OPC cases classified as HPV(+) expecting to find a similar association to the one using all OPC cases. Analyses restricting to HPV16 E6+ cases (>1000 MFI) captured again the association with rs4713462 (B = −0.1; $P = 0.01$). The association was less significant as expected given the smaller numbers and reduced range of data (Supplementary Fig. 11). We did not see association with the 71-Glu variant and L1 levels when restricting to the HPV16 L1+ cases (B = −0.04; $P = 0.22$). This could again be explained by the more limited data range when one restricts to L1+ cases (Supplementary Fig. 11).

Combined seropositivity to HPV16 E6 and L1 was relatively common among OPC cases (in 42%, 685 of 1643 OPC). We therefore investigated the possible influence of E6 or L1 antibody levels on their association with the respective genetic variant by adjusting for the alternative serology marker. As shown in Supplementary Table 11, results are extremely similar to the original results in Fig. 3, indicating that adjustment by the other serology marker makes little difference.

Due to the functional connection and correlation of other E proteins (E1, E2, E7) with HPV16 E6, we explored the relationship

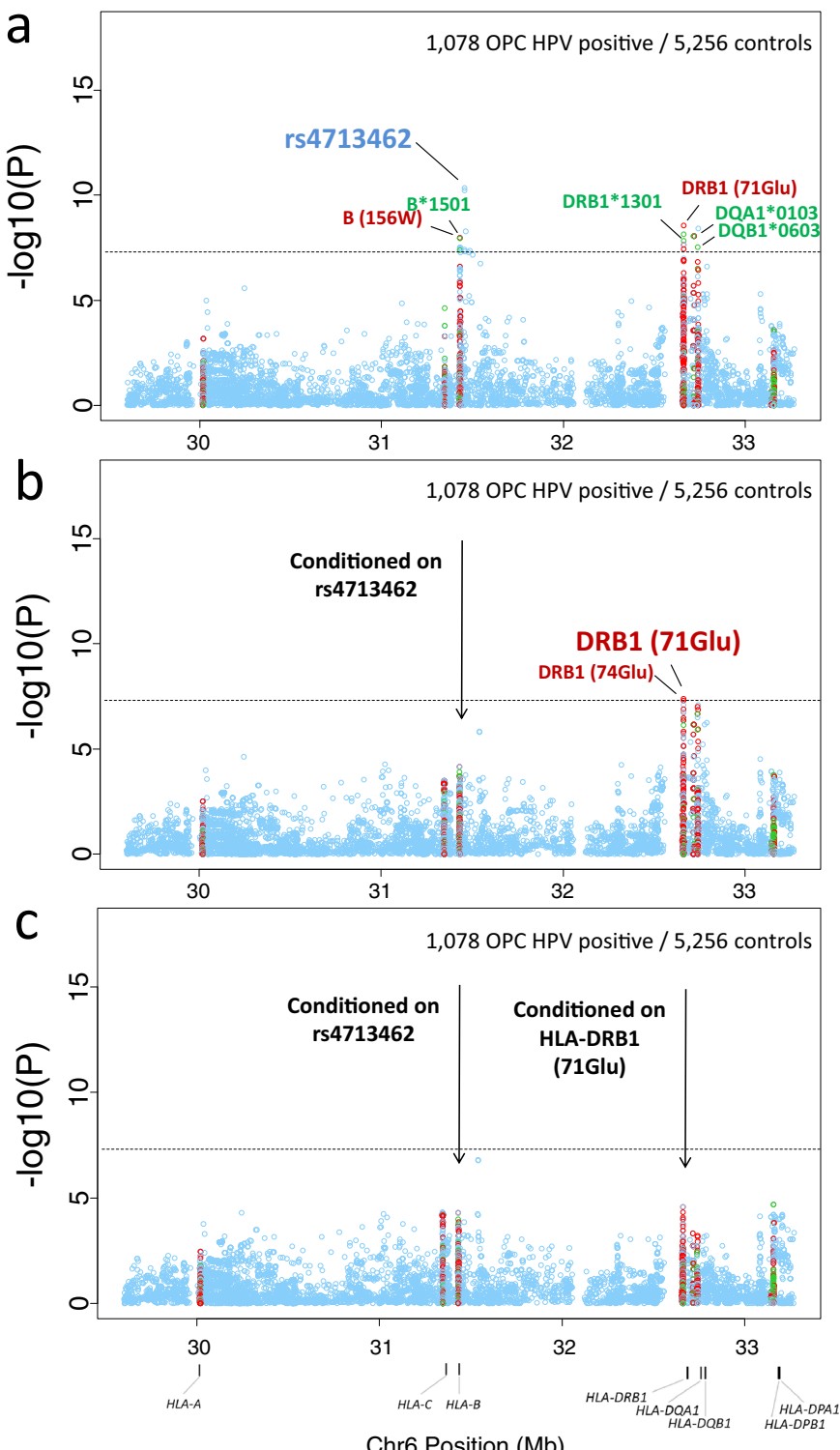

**Fig. 2 Plots of stepwise conditional association in the MHC region for HPV(+)OPC cases. a–c** The association for each locus used for conditioning is shown in each panel: **a** unconditioned, **b** conditioned on rs4713462, **c** conditioned on rs4713462 and HLA-DRB1 amino acid change in position 71 (71-Glu). Detailed association results in Table 2 and Supplementary Fig. 2. Circles represent –log10 (P values) for each binary marker using the imputed allelic dosage (between 0 and 2) and genotyped variants. Multivariable logistic regression assuming a log-additive genetic or dosage model with age, sex, and eigenvectors as covariates was used as baseline model. The dashed black horizontal lines represent the study-wide significant threshold of $P = 5 \times 10^{-8}$. The physical positions of HLA genes on chromosome 6 are shown at the bottom. The color of the circles indicates the type of marker; light blue—SNPs outside and within HLA genes, green - classical HLA alleles and red—amino acid polymorphisms of the HLA genes. HLA human leukocyte antigen, MHC Major Histocompatibility Complex, HPV human papillomavirus, OPC, oropharyngeal cancer.

**Table 3 HLA alleles, SNP and amino acids contained in the best models obtained from HPV(+)OPC analyses as judged by the BIC criterion.**

| Model | Locus | | OR (95%CI)[a] | P value | BIC | BIC dif[b] |
|---|---|---|---|---|---|---|
| A | rs4713462 + DRB1 (71-glu) | | | | 4454 | 0 |
| | | rs4713462 | 0.67 (0.59-0.76) | $1.46 \times 10^{-10}$ | | |
| | | DRB1 (71-glu)[c] | 0.55 (0.46-0.67) | $1.03 \times 10^{-9}$ | | |
| B | rs4713462 + Class II Haplotype | | | | 4465 | +11 |
| | | rs4713462 | 0.66 (0.58-0.75) | $5.52 \times 10^{-11}$ | | |
| | | DRB1*1301-DQA1*0103-DQB1*0603 | 0.45 (0.34-0.61) | $1.13 \times 10^{-7}$ | | |
| | HLA-B*1501 + Class II Haplotype | | | | 4489 | +35 |
| | | HLA-B*1501 | 0.53 (0.40-0.71) | $1.77 \times 10^{-05}$ | | |
| | | DRB1*1301-DQA1*0103-DQB1*0603 | 0.43 (0.32-0.58) | $3.23 \times 10^{-8}$ | | |
| | B (156) + Class II Haplotype | | | | 4489 | +35 |
| | | B (156) | 0.54 (0.41-0.71) | $1.53 \times 10^{-05}$ | | |
| | | DRB1*1301-DQA1*0103-DQB1*0603 | 0.43 (0.32-0.58) | $3.05 \times 10^{-8}$ | | |

*BIC* Bayesian information criterion, *HLA* human leukocyte antigen, *OR* odds ratio, *95% CI* confidence interval
[a]Obtained from multivariate logistic regression assuming an additive genetic model with sex and principal components as covariates.
[b]Models having their BIC difference within: +1–2 of the minimum have substantial support; +4–7 of the minimum have considerably less support; BIC > 10 above the minimum fail to explain some substantial structural variation in the data.
[c]Alleles where these amino acids are part of the sequence: 71(Glu) in *DRB1*\*1301 and also in \*0402, \*1302, and \*1102.

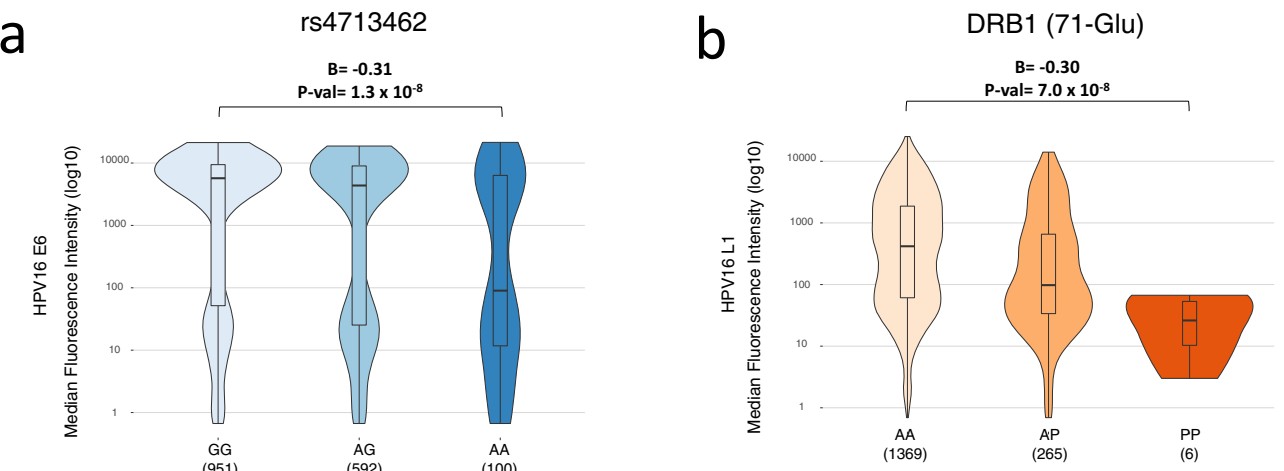

**Fig. 3 pQTL analysis of plasma antibody levels against HPV proteins of the top two associated HPV(+)OPC HLA variants.** Box plots showing **a** HPV16 E6 MFI levels and rs4713462 genotypes, and **b** HPV16 L1 MFI levels and DRB1 (71-Glu) variants in OPC cases. Effect sizes (B, regression coefficient) were adjusted for age, sex, and eigenvectors. For HLA-DRB1 71-Glu, A stands for absence and P stands for presence. Box plots show the medians (center lines) and the 25th and 75th percentiles (box edges), with whiskers extending to 1.5 times the interquartile range. Linear multivariate models were implemented to test associations of genetic variants across the MHC region with HPV16 L1 or E6 log-transformed MFI levels, assuming a dosage model with age, sex and eigenvectors as covariates. HLA human leukocyte antigen, pQTL protein Quantitative Trait Loci, HPV human papillomavirus, OPC oropharyngeal cancer, MFI median fluorescence intensity.

between the top associated HLA variants and antibodies against these proteins as a sensitivity analysis. We also explored association with other well-known but less common high-risk HPV types (E6 and E7 of HPV18/31/33/35/45/52/58) to assess the question of whether the identified HLA variants are E6 specific associated variants independent of HPV type. Our results suggest that the rs4713462 association might be specific to HPV16 E6 antibodies given that the strongest association was still the one detected with HPV16 E6 (Supplementary Fig. 6). Seropositivity for other HPV high-risk types represents a very limited fraction among HPV(+) or HPV(−)OPC reducing the chances for non-HPV16 markers to explain HLA genetic findings (Supplementary Tables 9 and 10).

**HPV(−)OPC GWAS.** In the GWA analyses within HPV(−)OPC cases and controls, a risk variant at 12p23.3 (rs35189640, OR = 2.73, $P = 1.1 \times 10^{-9}$; Fig. 1b; Table 2) was found significantly associated with risk of HPV(−)OPC. Genomic annotation revealed that rs35189640 is an intronic variant in the *BTB domain-containing 11* (*BTBD11*), a DNA-binding protein such as zinc-finger transcription factors. Interestingly, rs35189640 also maps into a repressed Polycomb-group protein (PcG) binding site specific to epithelial tissue based on chromatin states evaluated in more than 30 tissues (Supplementary Fig. 5). No evidence for heterogeneity was detected by geographic region or study characteristics displayed in Table 1 (Supplementary Fig. 2d). Other suggestive susceptibility variants were found at five additional loci at 3p26.2, 1p36.13, 13q33.3, 5q34, and 6p21.32 that fell just

under the GWA significance level ($P < 1 \times 10^{-7}$, Supplementary Table 6).

**Pooled GWAs of OCC and HPV(−)OPC cases**. To improve power and test the reproducibility of our findings, we pooled 2359 oral cavity cancer (OCC) cases with the 565 HPV(−)OPC cases and repeated the analyses. This was done under the assumption that OCC is a predominately HPV(−) disease and has risk factors in common with HPV(−)OPC, i.e., tobacco and alcohol usage.

Manhattan plots for the pooled dataset are displayed in Supplementary Fig. 7. In addition to previously reported variants[9] for both cancer sites, we identified two susceptibility loci at 6p21.32 (rs3828805, OR = 0.77, $P = 2.5 \times 10^{-9}$) and 15q21.2 (rs10851478, OR = 1.22, $P = 1.3 \times 10^{-8}$) common to both cancer entities. The rs3828805 is in LD with *HLA-DRB1* 71-Glu variant ($r^2 = 0.48$) that we found to be associated with HPV16 L1 antibody levels in HPV(+) OPC. As expected, rs4713462, which was found exclusively associated with antibodies against the HPV16 E6 viral oncoprotein and is independent from rs3828805 (r2 = 0.0004), was not associated with risk of OC/OPC(−) (Table 4). Regarding rs10851478, this variant is a *cis*-eQTL for the *FGF7* gene (Supplementary Fig. 8) which is a potent epithelial cell-specific growth factor with an important role in morphogenesis of epithelium.

**Assessment of the consistency of effects across studies**. Overall, heterogeneity tests showed that HPV(−)OPC and OC share their top disease-related genetic associations while those for HPV(+) and HPV(−)OPC were different revealing a different genetic predisposition and probably genetic architecture (Supplementary Fig. 10). In brief, significant heterogeneity ($p < 0.05$) was identified in the meta-analysis of HPV(+) and HPV(−)OPC for rs4713462 at HLA region and for rs35189640 at *BTBD11* locus suggesting an exclusive association with HPV(+) and HPV(−) OPC, respectively (Supplementary Fig. 10b). *FGF7* and *LHPP* association showed common and consistent effects to the HPV(-) OPC and OCC analyses (Supplementary Fig. 10a) as well as *ADH1B* that was previously identified as associated with overall OPC and OCC[9]. *BTBD11* showed a stronger effect for HPV(−) OPC (Supplementary Fig. 10).

## Discussion

Overall, our study identified OPC genetic heterogeneity by HPV16 status and expands existing knowledge of the genetics of oral cancers identifying several genetic loci.

Two loci in the HLA region showed the strongest protective effects against HPV(+)OPC risk: rs4713462 and an amino acid polymorphism in *HLA-DRB1* 71-Glu. The latter refines the previously reported HLA class II haplotype association recently replicated in an independent series[9,18]. Using HPV serology data, rs4713462 was exclusively associated with antibody levels against HPV16 E6, the viral oncoprotein, while *HLA-DRB1* 71-Glu was only associated with antibody levels against HPV16 L1, the viral capsid protein.

In our current understanding, HPV seropositivity is the result of a humoral immune response to a viral protein in those individuals trying to mount a T-cell response to eliminate affected cells[19,20]. T-cell-mediated immune responses are essential for the clearance of both HPV infection and HPV-driven tumor cells and one may speculate that the associated HLA variants are promoting a targeted cell-mediated response to viral proteins which confers the protective effect against an HPV-driven cancer. Indeed, the relatively lower MFI levels of L1 or E6 in those individuals carrying the *HLA-DRB1* 71-Glu or the A allele of rs4713462 respectively may be explained by more active "consumption" of antibodies when the T cell sees its antigen reflecting an active immune reaction[21,22]. Consequently, the *HLA-DRB1* 71-Glu variant may implicate a more effective control of preventing an HPV16 infection that ultimately protects against HPV-driven cancers. This would be mimicking the effect of prophylactic vaccines which are exploiting the role of L1 capsid protein in initiating a tissue-specific infection and its antigenicity or maybe inducing a more effective T-cell response against L1 and thus preventing oncogenic transformation. The A allele of rs4713462 may be acting at a later stage through a targeted cell-mediated response to viral carcinogenesis provided by HPV16-infected cells (over)expressing E6. Therefore, our results provide evidence of natural immune response promoted by HLA germline variants against HPV-driven tumors at different stages of the infectious process that leads to cancer. This suggests, in the case of the E6-related variant, that a therapeutic vaccine targeting HPV16 E6 may attenuate viral impact after established HPV infections[23]. Several promising approaches have been developed, with some having achieved advanced clinical trial stages[24–27]. Oncoproteins E6 and E7 have been widely used in therapeutic HPV vaccine studies, given that the HPV16 E6 and E7 proteins are both drivers of unchecked cell proliferation in the most common oncogenic HPV types and are constitutively expressed. HPV vaccines using E7 alone have been more commonly used than E6. The fact that E7 is more abundantly expressed and highly conserved makes it a priori a better candidate, but it is a poorly immunogenic protein in reality. Our results put a particular emphasis on the potential of protein/peptide vaccines against E6. However, combination with other therapies targeting regulatory mechanisms and local immunosuppression in the tumor microenvironment, might be necessary for cancer therapeutic vaccination to become clinically successful.

**Table 4 Top genome-wide significant regions and the highlighted ones in previous analysis from regional meta-analyses of OCC and HPV(-)OPC pooled analysis.**

| Genomic regions | Locus | SNP | Info[a] | EA/OA[b] | Meta-analysis[c] | | |
|---|---|---|---|---|---|---|---|
| | | | | | OR (95%CI) | *P* value | Q_p |
| 6p21.32 | HLA region | rs3828805 | 0.88 | T/C | 0.77 (0.71–0.83) | $2.5 \times 10^{-9}$ | 0.71 |
| | | rs4713462 | Genotyped | A/G | 1.02 (0.95–1.10) | 0.61 | 0.10 |
| 4q23 | *ADH1B* | rs1229984 | Genotyped | A/G | 0.59 (0.49–0.71) | $4.1 \times 10^{-8}$ | 0.11 |
| 10q26 | *LHPP* | rs201982221 | Genotyped | D/I | 1.74 (1.43–2.13) | $4.8 \times 10^{-8}$ | 0.28 |
| 12p23.3 | *BTBD11* | rs35189640 | 0.94 | T/C | 1.79 (1.45–2.21) | $7.5 \times 10^{-8}$ | 0.54 |
| 15q21.2 | *FGF7* | rs12910284 | 0.99 | G/A | 1.22 (1.14–1.30) | $1.5 \times 10^{-8}$ | 0.18 |

*OR odds ratio, 95% CI confidence interval, Q_p Cochran's heterogeneity p value.*
[a]Imputation INFO (R2) is the average score across imputation batches.
[b]Effect allele/other allele; "D" stands for deletion and "I" stands for insertion.
[c]Regional meta-analyses Europe and North America with a total number of subjects: 2923 patients and 5256 controls.

The HPV(−)OPC analysis revealed one association at 12p23.3 in *BTBD11* and showed shared genetic links with OCC identifying two risk loci in addition to those that were already known: 6p21.32 and 15q21.2. Functional evidence at 12p23.3 was limited, even though we found that the *BTBD11* variant maps into a tissue-specific PcG protein binding site and nearby a transcription starting site. This suggests a potential impairment of transcription regulation increasing risk for epithelial malignancies[28–30]. At 15q21.2, rs10851478 showed a significant risk effect to develop HPV(−) oral cavity cancers and also tend to be a cis-eQTL for the *FGF7* gene specific of fibroblasts. *FGF7* is a potent epithelial cell-specific growth factor with an important role in morphogenesis of epithelium suggesting this variant as one of the malignancy-contributing factors from tumor stroma[31,32].

A limitation of the current study is the relatively modest sample size for some of the analyses. However, the inclusion of a rich collection of data partially compensates this limitation. For HPV(+)OPC, convergent evidence of association from distinct approaches (GWAs and pQTL) identify the same variants within HLA as the best associated variants in relation with HPV infection. For HPV(-) cancer the statistical power conferred by using a second dataset with potential genetic links has allowed us to support the discovery of additional genome-wide significant SNPs.

To conclude, the results generated in this study are important for understanding the biology of oral and OPC. Although future functional analyses are needed to confirm the impact of our findings, we provide extensive information on the likely consequences of relevant genetic variants putting forward a rich set of plausible gene targets and biological mechanisms for functional follow-up for both HPV(+)OPC and HPV(−)OPC and OCC cancers. In particular, our findings enhance the role of HLA variants in the immune pathogenesis of HPV(+)OPC, and may also have implications for cancer immunotherapies such as HPV-targeted vaccines that are currently undergoing clinical evaluation.

## Methods

**Subjects, specimens, and genotypes.** A total of 1643 OPC cases (included ICD codes: C01.9, C02.4, and C09.0–C10.9) and 5256 controls were included in this study (Table 1). All study participants were of European ancestry and came from 10 epidemiological studies from North America and Europe included in the so-called OncoArray study published earlier in 2016[9] (Supplementary Table 1). Subjects were genotyped using the Illumina OncoArray. Plasma samples utilized for HPV status determination were obtained from HPV unvaccinated subjects at time of diagnosis and prior to start of treatment for OPC cases, and at time of enrollment for controls.

The Oncoarray study also includes OCC (included ICD codes: C02.0–C02.9 (except C02.4), C03.0–C03.9, C04.0–C04.9, C05.0–C06.9). In contrast to OPC, HPV is thought to play a very limited role in OCC, with an estimated prevalence of 5% or less in most populations[33,34]. In this regard, OC shares risk factors with HPV(−)OPC, namely tobacco and alcohol usage. We therefore hypothesized that HPV(−)OPC and OCC may share genetic risk factors and, to expand the study population for HPV(−) tumors, pooled together the series of 2,359 OCC cases and 565 HPV(−)OPC and performed a GWA analysis (Supplementary Table 4).

**Ethics statement.** Each included study was approved by their corresponding local ethics committee/Institutional Review Board, and informed consent was obtained from all study participants to undertake serological analyses which were not covered in our previous study[9]. In addition, the International Agency for Research on Cancer Institutional Review Board (IARC-IRB; reference 16–34) evaluated and approved inclusion of each contributing study into the current study

**Imputation and HLA fine mapping.** For all subjects, individual-level genome-wide SNP data were available[9]. Imputation was performed using the Michigan Imputation Server[35] with the Haplotype Reference Consortium panel[36]. SNPs with minor allele frequency (MAF) < 0.01 or $R^2$ < 0.3 in any of the batches were excluded before association analyses. Taking advantage of the high OncoArray SNP coverage in the HLA region, we used the genotyping data from 25 to 35 Kb at chromosome 6 (NCBI build 37) to impute 8,961 HLA variants (including classical two and four digit HLA alleles and amino acid polymorphisms of the HLA genes

along with the SNPs across MHC region) that were not directly genotyped. For this, we utilized the reference data collected by the Type 1 Diabetes Genetics Consortium (T1DGC)[37,38] using the SNP2HLA v1.0.3 package in Beagle software v3[39,40]. We applied post-imputation QC criteria of $R^2$ < 0.3 to exclude variants from the association analysis. All the evaluated HLA variants were defined as binary markers in the analyses as follows: (a) For biallelic SNPs, classical HLA alleles and binary amino acid positions, the effect allele or variant was the minor allele, the presence of the HLA allele or the presence of the less frequent amino acid, respectively; (b) For multi-allelic amino acid positions, composite markers were defined where each possible individual allele and combination of alleles was tested for association.

**Serological analyses and HPV status determination.** Plasma samples were analyzed at the German Cancer Research Center (DKFZ; Heidelberg, Germany) using multiplex serology by laboratory staff blinded to the case–control status and cancer site of the subjects[41]. Briefly, viral antigens were affinity-purified, bacterially expressed fusion proteins with N-terminal Glutathione S-transferase. Biotinylated secondary antibodies (Jackson ImmunoResearch Laboratories, reference: 109-065-064) were used in a 1:1000 dilution as reporters to detect bound plasma antibodies. Samples were evaluated at a final 1:100 dilution for antibodies against the HPV16 oncoproteins E6 and E7, other viral regulatory proteins (E1, E2), and the major capsid protein L1 as well as for antibodies against proteins from other high-risk HPV types (HPV18/31/33/35/45/52/58) (Supplementary Table 5). MFI values were dichotomized as antibody positive or negative for HPV16 proteins using predefined cutoff values, that is 1000 MFI (E6), 548 MFI (E7), 200 MFI (E1), 679 MFI (E2) and 422 MFI (L1)[6,14]. OPC cases were classified as HPV(+) or HPV(−) based on a previously validated HPV16 seropattern algorithm[14] with HPV(+) defined as follows: HPV16 E6 > 1000 or 3 out of 4 E proteins greater than their cutoff value (HPV16 E1 > 200 MFI, HPV16 E2 > 679 MFI, HPV16 E6 > 484 MFI, HPV16 E7 > 548 MFI). Subjects who did not fulfill the criteria for HPV(+) were classified as HPV(−).

**GWAS, regional meta-analyses.** Regional meta-analysis (Europe and North America) was performed to evaluate the relationship between SNPs and OPC risk by HPV status and between SNPs and risk of OCC or HPV(−)OPC (pooled series of OCC and HPV(−)OPC). In brief, GWAs for each world region were performed for genotype dosages, using multivariable unconditional logistic regression assuming a log-additive genetic or dosage model with age, sex, and eigenvectors as covariates. Next, association statistics were included in a fixed-effect meta-analysis performed in PLINK v1.07[42]. The P value for heterogeneity was calculated using Cochran's Q test. Conditional analyses within associated regions and meta-analysis of regional results were performed using glm funtion in R software[43] (v.3.6.3).

**Assessment of the consistency of effects across studies.** A test for heterogeneity (Cochran's Q) was used to evaluate the consistency of the effects of the top hits across GWAs by combining in a meta-analysis HPV(+) and HPV(−)OPC results or HPV(−)OPC and OC after dividing the control group into two equivalent random series.

**Analysis within the MHC region.** Phased genotypes obtained from the imputed data were used to generate population-based haplotypes with the Haplo.stats R package (v.1.7.7).

Each haplotype, allele, variant, and associated amino acid was individually tested for an association with risk of OPC by HPV status within the statistical framework and covariates described previously (age, sex, and eigenvectors). Variables that were independently associated with risk (HLA alleles and/or haplotypes, their significant amino acids and SNPs) were assessed in multivariable risk models to search for the best overall HLA model. First, we used a forward selection stepwise regression of significant variants within the independently associated regions. For each gene or region we looked for the set of significant amino acid positions and we defined the classical alleles with consistent residues at those positions. We also included as covariates in these risk models the top independently associated markers identified within the HLA region from multivariable regression analyses. Model selection for the best overall HLA predictors associated with risk was conducted using the lowest BIC[44,45].

The aim was to find the best set of amino acids and/or HLA alleles that were independently associated with lung cancer as judged by the lowest AIC (Akaike Information Criterion) and BIC.

Three-dimensional HLA ribbon models for the HLA-B and HLA-DR proteins were prepared using UCSF Chimera v1.11.

**Functional variant annotation.** To identify potentially functional candidates a comprehensive bioinformatic assessment of the SNPs with GWAS P value < $10^{−6}$ was performed. Using the HaploR R package, both HaploReg[46] and RegulomeDB[47] were crosschecked to explore data reflecting transcription factor binding, open chromatin and the presence of putative enhancers. Further, the Genotype-Tissue Expression dataset (GTEx v8)[48] was interrogated to identify potential genes influenced by disease-associated SNPs. For this purpose, GTEx v8 eQTL summary statistics, based on RNAseq analysis, were obtained for tissues best representing

oropharyngeal and oral cavity tissue in this platform (Esophagus mucosa of 497 individuals, Esophagus muscularis of 465 individuals, not sun-exposed skin of 517 individuals and cell-cultured fibroblasts of 483 individuals). Functional annotation for variants reaching $P < 5 \times 10^{-8}$ is summarized in Supplementary Table 7.

Linkage Desequilibrium LD statistics (D′ and R²) of genomic regions of interest were also examined in the reference European population from Phase 3 of the 1000 Genomes Project (1000G)[49].

To evaluate potential overlap with published GWAS findings, results reported in the NHGRI catalog[50] were extracted on 2020-12-02 (containing 4795 publications and 222481 associations) and crosschecked against our results.

**Protein quantitative trait loci (pQTL) of plasma antibodies levels against HPV proteins.** As mentioned, plasma antibody levels against HPV16 L1 are thought to represent cumulative past and present HPV infection from multiple possible anatomic sites[11,13,51–53] while E6 antibody levels are accurate markers of malignant HPV-related disease[14,16,54]. Considering the connection between antibody-mediated immune responses and HLA-mediated antigen recognition, we explored the impact of MHC genetic variants on the humoral response to HPV proteins on the whole series of OPC cases. Linear multivariate models were implemented to test associations of genetic variants across the MHC region with HPV16 L1 or E6 log-transformed MFI levels, assuming a dosage model with age, sex, and eigenvectors as covariates in OPC cases.

**Technical validation of imputed SNPs.** Technical validation of imputed findings was performed in the recent GWAS study[9]. In addition, the HPV(−)OPC-associated low-frequency variant at 12q23.3 (rs35189640) was successfully validated by TaqMan assays (Thermo Fisher Scientific) in 1810 individuals (concordance of 99.9%).

**Reporting summary.** Further information on research design is available in the Nature Research Reporting Summary linked to this article.

## Data availability

The authors declare that all data supporting the findings of this study are provided within the paper and its supplementary information. Genotype data for the OPC and OC OncoArray have been deposited at the database of Genotypes and Phenotypes (dbGaP) and are available under controlled access under accession phs001202.v1.p1. All the projects are generally approved for non-commercial research and we ensure that the proposed research complies with the signed agreements with research participants. The oral and pharyngeal GWAS summary statistics by cancer site and world region have been deposited in the IEU Open GWAS platform (https://gwas.mrcieu.ac.uk/) under the GWAs IDs: ieu-b-89, ieu-b-90, ieu-b-94, ieu-b-96, ieu-b-93, ieu-b-91, ieu-b-95 and ieu-b-98. Protein Data Bank (RCSB PDB) structures for three-dimensional HLA ribbon models can be obtain under accession codes 3pdo and 2bpv. Genotype-Tissue Expression dataset (GTEx v8) is publicly available and can be downloaded following instructions at [https://gtexportal.org/home/protectedDataAccess].

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

## Acknowledgements

This study is part of an NIDCR funded project called VOYAGER (R01DE025712) with co-PIs Paul Brennan (IARC, Lyon), Brenda Diergaarde (University of Pittsburgh) and Neil Hayes (University of Tennessee Health Science Centre). Genotyping of cases and controls was performed at the Center for Inherited Disease Research (CIDR) and funded by the US National Institute of Dental and Craniofacial Research (NIDCR; 1X01HG007780-0). The work performed in MSH-PMH study was supported by The Canadian Cancer Society Research Institute (020214), Ontario Institute of Cancer, Canada Research Chair Award to R.J.H. and the Alan Brown Chair to G.L. and Lusi Wong Programs at the Princess Margaret Hospital Foundation. Coordination of the EPIC study is financially supported by the European Commission (DG SANCO) and the International Agency for Research on Cancer. The University of Pittsburgh head and neck cancer case–control study is supported by US National Institutes of Health grants P50CA097190 and P30CA047904. The Alcohol-Related Cancers and Genetic Suscep-tibility Study in Europe (ARCAGE) was funded by the European Commission's fifth framework program (QLK1-2001-00182), the Italian Association for Cancer Research, Compagnia di San Paolo/FIRMS, Region Piemonte and Padova University (CPDA057222). The Carolina Head and Neck Cancer Study (CHANCE) was supported by the National Cancer Institute (R01CA90731). This publication presents data from the Head and Neck 5000 study. The study was a component of independent research funded by the National Institute for Health Research (NIHR) under its Program Grants for Applied Research scheme (RP-PG-0707-10034). The views expressed in this publication are those of the author(s) and not necessarily those of the NHS, the NIHR or the Department of Health. Core funding was also provided through awards from Above and Beyond, University Hospitals Bristol Research Capability Funding and the NIHR Senior Investigator award to Professor Andy Ness. Human papillomavirus (HPV) serology was supported by a Cancer Research UK Programme Grant, the Integrative Cancer Epide-miology Programme (grant number: C18281/A19169). This study is supported by the US National Institute of Dental and Craniofacial Research (R01DE025712 to P.B., B.D., and N.H.). The IARC Oral Cancer Multicenter study was funded by grant S06 96 202489 05F02 from Europe against Cancer; grants FIS 97/0024, FIS 97/0662 and BAE 01/5013 from Fondo de Investigaciones Sanitarias, Spain; the UICC Yamagiwa-Yoshida Memorial International Cancer Study; the National Cancer Institute of Canada; Associazione Italiana per la Ricerca sul Cancro; and the Pan-American Health Organization. The IARC Central Europe study was supported by the European Commission's INCO-COPERNICUS Program (IC15-CT98-0332), US NIH/National Cancer Institute grant CA92039 and World Cancer Research Foundation grant WCRF 99A28. The Rome Study was supported by the Associazione Italiana per la Ricerca sul Cancro (AIRC) awards IG 2011 10491 and IG 2013 14220 to S.B. and by Fondazione Veronesi to S.B. This pub-lication presents data that contributes to the HEADSpAcE study supported by the European Union Horizon 2020 (grant no 825771).

## Author contributions

Formal analyses: A.F.I., C.L., V.G., NicoleB, V.D.-O., M.M. HPV serology: T.W., LeaS, NicoleB, N.B., MichaelP. Oropharyngeal and Oral OncoArray project coordination: C.I.A., J.D.M., R.J.H., P.B. Contributed with reagents, samples, data harmonization, and/ or materials: A.A., A.Z., M.N., B.T.H., C.M.H., D.I.C., F.M., I.H, J.P., A.L., L.S., M.R., P.L., T.V.M., W.A., A.F.O., M.C.W., N.H., M.J.J., A.R.N., M.P., S.T., J.D.M., S.W., S.V., D.M., K.S.B., B.K., P.B., M.L., R.H., C.I.A., B.D., R.F., S.B., G.C., L.A., G.L., R.J.H., SeraT, Y.B., T.D. Project coordination: P.B., B.D., N.H., T.W., A.N., R.J.H., G.L., A.F.O. J.D.M. Conceptualization: A.F.I., P.B., T.W., A.N., J.D.M., B.D., N.H., R.J.H., G.L. A.F.O. Orig-inal Draft Preparation: A.F.I., P.B. All co-authors made substantial contributions to data acquisition, data interpretation, and revised the work critically for important intellectual content

## Competing interests

R.F. has the following financial disclosures: Aduro Biotech, Inc (consulting); Astra-Zeneca/MedImmune (clinical trial, research funding); Bristol-Myers Squibb (advisory board, clinical trial, research funding); EMD Serono (advisory board); MacroGenics, Inc (advisory board); Merck (advisory board, clinical trial); Novasenta (consulting, stock, research funding); Numab Therapeutics AG (advisory board); Pfizer (advisory board); Sanofi (consultant); Tesaro (research funding) and Zymeworks, Inc (consultant). Where authors are identified as personnel of the International Agency for Research on Cancer/ World Health Organization, the authors alone are responsible for the views expressed in this article and they do not necessarily represent the decisions, policy or views of the International Agency for Research on Cancer/World Health Organization. All other authors have no conflicts to disclose.

## Additional information

[1]Section of Genetics, Genetic Epidemiology Group, International Agency for Research on Cancer, World Health Organization, Lyon, France. [2]Section of Genetics, Genetic Cancer Susceptibility Group, International Agency for Research on Cancer, World Health Organization, Lyon, France. [3]Infections and Cancer Epidemiology, German Cancer Research Center (DKFZ), Heidelberg, Germany. [4]Department of Environmental

Medicine and Public Health, Icahn School of Medicine at Mount Sinai, New York, NY, USA. [5]National Institute for Health Research (NIHR) Bristol Biomedical Research Centre, University Hospitals Bristol and Weston NHS Foundation Trust and the University of Bristol, Bristol, UK. [6]Bristol Dental School, University of Bristol, Bristol, UK. [7]Prosserman Centre for Population Health Research, Lunenfeld-Tanenbaum Research Institute, Sinai Health System, Toronto, ON, Canada. [8]Lunenfeld-Tanenbaum Research Institute of Sinai Health System, University of Toronto, Toronto, ON, Canada. [9]Department of Human Genetics, Graduate School of Public Health, University of Pittsburgh, Pittsburgh, PA, USA. [10]UPMC Hillman Cancer Center, Pittsburgh, PA, USA. [11]Department of Epidemiology, Gillings School of Global Public Health, University of North Carolina at Chapel Hill, Chapel Hill, NC, USA. [12]Division of Medical Oncology and Center for Cancer Research, University of Tennessee Health Science Center, Memphis, TN, USA. [13]Department of Otolaryngology/Head and Neck Surgery, University of North Carolina at Chapel Hill, Chapel Hill, Chapel Hill, NC, USA. [14]Oncology Data Analytics Program, Catalan Institute of Oncology (ICO), Barcelona, Spain. [15]Section of Early Detection and Prevention, Prevention and Implementation Group, International Agency for Research on Cancer, World Health Organization, Lyon, France. [16]Section of Hygiene, University Department of Life Sciences and Public Health, Università Cattolica del Sacro Cuore, Roma, Italy. [17]Department of Woman and Child Health and Public Health - Public Health Area, Fondazione Policlinico Universitario A.Gemelli IRCCS, Roma, Italy. [18]Department of Head and Neck Surgery, Institute of Clinical Otorhinolaryngology, Università Cattolica del Sacro Cuore, Roma, Italy. [19]Istituto di Clinica Otorinolaringoiatrica, Fondazione Policlinico Universitario A.Gemelli IRCCS, Roma, Italy. [20]Department of Otorhinolaryngology, Head and Neck Surgery, Maastricht University Medical Center, Maastricht, The Netherlands. [21]Institute of Hygiene and Epidemiology, Prague, Czech Republic. [22]University Bremen, Bremen, Germany. [23]School of Medicine, National and Kapodistrian University of Athens, Athens, Greece. [24]School of Public Health, University of West Attica, Athens, Greece. [25]National Cancer Institute, IRCCS, Aviano, Italy. [26]University of Padova, Padova, Italy. [27]CeRMS and University of Turin, Turin, Italy. [28]Trinity College School of Dental Science, Dublin, Ireland. [29]Cancer Registry of Norway, Oslo, Norway. [30]Department of Research, Cancer Registry of Norway, Oslo, Norway. [31]School of Medicine, Dentistry, and Nursing, University of Glasgow, Glasgow, UK. [32]Queen Elizabeth University Hospital, NHS Greater Glasgow and Clyde, Glasgow, UK. [33]University of Aberdeen, Aberdeen, UK. [34]Centre for Oral Health Research, Newcastle University, Newcastle, UK. [35]Catalan Institute of Oncology/IDIBELL, Barcelona, Spain. [36]Centro de Investigación Biomédica en Red de Epidemiología y Salud Pública-CIBERESP, Madrid, Spain. [37]Cancer Surveillance Section, International Agency for Research on Cancer, World Health Organization, Lyon, France. [38]Department of Medicine, Baylor College of Medicine, Houston, TX, USA. [39]These authors jointly supervised this work: Tim Waterboer, Paul Brennan. ✉email: ferreiroa@iarc.fr; gep@iarc.fr

