## [Peer Review File · Nature Communications]

Germline Determinants of Humoral Immune Response To HPV-16 Protect Against Oropharyngeal CancerReviewers' Comments:

Reviewer #1:

Remarks to the Author:

"Germline Determinants of Humoral Immune Response to HPV-16 Protect Against Oropharyngeal Cancer," by Ferreiro_Iglesias et al. determines the relationship between E6 seropositivity and OPSCC and germ line variation.

Because E6 seropositivity is believed to indicate HPV positivity in at least 90% of OPSCC cases the authors make the logical conclusion that it can be used to accurately identify HPV positive cases in their study group. It is good that the plasma for this measurement was taken at the time of diagnosis as that is how the methodology of identifying HPV positive tumors was originally validated for its >90% accuracy. Using this information the authors stratify OPSCC as E6+ (or HPV+) and E6- or HPV-. This is a large scale collaboration that few others can do and it is an important step forward in the characterization of HPV linked oral pharyngeal SCC carcinogenesis. I support acceptance with some revisions.

It needs to be clearly mentioned in the introduction or results, not just in the methods section/supplement, that the study set is a subset of those published earlier in Nat. Genet. 2016 article from some of the same workers. Given the time of sample collection it is probably a low number, but if there are an appreciable number of subjects who received prophylactic L1 vaccine prior to diagnosis, presumably mainly in the negative controls, then that needs to be reported. This may be somewhere in the manuscript but I could not find it.

There is a bit of a problem using an immune factor to identify genetic variability at an immune associated HLA site. For example, what if the rs4713462 allele (A) deficit is that the subject is resistant to raising an immune response to E6, and becoming E6 seropositive, then a result similar to what is seen here would occur. Individuals with that locus would be less frequently E6 seropositive but it would not be because they have fewer HPV positive tumors. Do the authors have a way to rule that out? I am not saying it is probable but it is possible. These might show up as HPV L1 positive OPSCCs that are E6 negative.

It would be interesting to know if the rs4713462 and rs3828805 loci variants show the same association with HPV positivity in patients without OPSCC that they do in the OPSCC patients. I realize the numbers are too low to do that. Based on their data the authors might make their conclusions clearer if they make a statement about what they would predict in that situation..

The authors' pooling of oral pharyngeal SCC HPV- and oral SCC makes sense and it is gratifying the two groups share similar disease-related genetic associations.

The authors speculation on the possible efficacy of developing a OPSCC treatment with an E6 vaccine is interesting.

Line 115. serology is an ideal solution in situation where tumor tissue for direct assessment is not available.

Please remove the word "ideal". Maybe replace with "a good"

Reviewer #2:

Remarks to the Author:

This manuscript aims to determine genetic associations of host genetic polymorphisms with

oropharyngeal cancer, through a large case-control GWAS study of HPV+ patients. GWAS significantly identified a signal nearby HLA-B and another signal pointing to the residue 71 on DRB1, in addition to two other signals on chromosome 3 and 4. Interestingly, the associations of HLA class I and II genes associate with the protection against OPC among HPV positive cases, suggesting potentially important protective roles of host immunity mediated through HLA genes. One major concern is with inadequate explorations of multiple genetic variants arising from this highly polymorphic MHC region in which linkage-disequilibrium (LD) is extensive. For example, it is not clear whether or not rs4713462 SNP is high LD with any allele(s) of HLA-B given its proximity, and further it is not clear if the single residue beta71 of DRB1 would be a critical residue contributing to the OPC susceptibility. It would be important to dig deeper into all genetic variants at nucleotide, residue, and gene allele levels. For such explorations, the conditional analysis is conventionally used, to adjust one risk factor when exploring other factors. However, one should be mindful of its intrinsic limitation, i.e., empirically adjusted factors through a conditional analysis can still have a causal association of importance. Here are specific concerns:

- 1) Page 4 and Line 110: It is advisable to adopt the official HLA nomenclature for the haplotype DRB1*13:01-DQA1*01:03-DQB1*06:03
- 2) P6L157: Define MFI
- 3) P6L169: It seems that two loci on chromosome 3 and 4 are found in the marginal association. It is equally informative to assess these genetic associations, after adjusting for HLA genes.
- 4) P7L185: Given the novel finding of this SNP rs4713462, it seems important to explore its LD relationship with HLA-B and -C, given its proximity. This exploration becomes particularly important, since this SNP is found to be a eQTL with HLA-C.
- 5) P7L197-L198: It is not clear how you are able to map multiple nucleotide associations (Supplementary Table 13) to a single residue beta71. Given extensive LD across DRB1-DQA1-DQB1, it is expected that multiple residues may be in high LD with beta 71.
- 6) P8L221-222: BIC is commonly used to select variables to build prediction models. In the selection process, the analysis procedure identifies those variables that have overall stronger "empirical associations", without regarding to underlying biological (causal) associations. Hence, one needs to be cautiously interpreting selected variants, amino acids and haplotypes.
- 7) P306L306-307: It is not clear if rs4713462 is accounted for by one or more HLA-B and -C alleles/haplotypes, nor if the single residue beta71 is responsible for HLA-DRB1 association
- 8) P12L312-314: HPV seropositivity is indicative of a humoral responses (humoral immunity), which is somewhat different from the T cell-mediated immunity. However, given identified genetic associations with class I and class II genes, it is plausible both humoral and cell immunities are involved in the protection of OPC. It may be helpful to expand the Discussion.
- 9) P12L322-324: It would be very helpful to expand discussion on this tantalizing possibility of developing prophylactic vaccines given role of L1 capsid proteins.
- 10) P16L419-426: It would be informative to carry out "formal haplotype association analysis" of HLA-A, HLA-B/C, HLA-DRB1, HLA-DQA1/B1, with imputed HLA genotypes, with or without other SNPs. If results are informative, they can be included, given their importance. Even if they are secondary, it would be helpful to include them in supplementary tables.

REVIEWER COMMENTS

Reviewer #1, expert in oral cavity and nasopharyngeal cancer (Remarks to the Author):

"Germline Determinants of Humoral Immune Response to HPV-16 Protect Against Oropharyngeal Cancer," by Ferreiro_Iglesias et al. determines the relationship between E6 seropositivity and OPSCC and germ line variation.

Because E6 seropositivity is believed to indicate HPV positivity in at least 90% of OPSCC cases the authors make the logical conclusion that it can be used to accurately identify HPV positive cases in their study group. It is good that the plasma for this measurement was taken at the time of diagnosis as that is how the methodology of identifying HPV positive tumors was originally validated for its >90% accuracy. Using this information, the authors stratify OPSCC as E6+ (or HPV+) and E6- or HPV-. This is a large-scale collaboration that few others can do and it is an important step forward in the characterization of HPV linked oral pharyngeal SCC carcinogenesis. I support acceptance with some revisions.

It needs to be clearly mentioned in the introduction or results, not just in the methods section/supplement, that the study set is a subset of those published earlier in Nat. Genet. 2016 article from some of the same workers. Given the time of sample collection it is probably a low number, but if there are an appreciable number of subjects who received prophylactic L1 vaccine prior to diagnosis, presumably mainly in the negative controls, then that needs to be reported. This may be somewhere in the manuscript, but I could not find it.

We thank the reviewer for this point. It is however unlikely that any subjects would have received a HPV vaccination prior recruitment. HPV vaccination was introduced in North America and Europe in 2006–2007, and was originally recommended for teenage girls, and only in the last years broadened to include boys. Our collection is composed of approximately 70% of men and recruitment period across studies ranged overall from 2002 to 2016 with a median age at enrolment of 58 years (IQR; 52 – 65 years) for cases and 62 years (IQR; 54 – 71 years) for controls. Given this, it is extremely unlikely that any vaccinated person was subsequently included as a control. We added a note clarifying this point in Methods section (line 363, page 14).

As requested by the reviewer we have also stated in methods and in the introduction that this work uses a subset of the data included in our earlier publication in Nature Genetics (PMID: 27749845) instead of only citing it. Methods section (line 360-361, page 14) and Introduction (line 108, page 4).

There is a bit of a problem using an immune factor to identify genetic variability at an immune associated HLA site. For example, what if the rs4713462 allele (A) deficit is that the subject is resistant to raising an immune response to E6, and becoming E6 seropositive, then a result similar to what is seen here would occur. Individuals with that locus would be less frequently E6 seropositive but it would not be because they have fewer HPV positive tumors. Do the authors have a way to rule that out? I am not saying it is probable, but it is possible. These might show up as HPV L1 positive OPSCCs that are E6 negative.

Thank you for this interesting comment. We agree with the reviewer that resistance to mount a humoral immune response against E6 in subjects with a HPV16-driven oropharynx tumor is conceivable, although is likely to be a rare situation. Indeed, the concordance of E6 seropositivity with tumor status has been estimated to be as high as >90%, indicating that very few cases with a negative E6 serology but a HPV16 positive OPC tumor are expected. Since L1 seropositivity is primarily a marker of past infection and does not represent an accurate marker of tumor status regardless of E6 serostatus (Holzinger et al.2017, PMID: 28316084), we differ with the reviewer suggestion as L1 seropositive OPSCCs will not be capturing accurately those HPV16-driven cancers. In

our data we have identified only 70 out of 1643 OPC cases fitting with the scenario suggested (L1 seropositive and E6 seronegative). Unfortunately, HPV tumor status is not available for these subjects and we cannot verify if there is any HPV positive tumor within this group. However, if this were the case, we would expect a deficiency of rs4713462 allele (A) but with no deviation from Hardy-Weinberg Equilibrium was found ($p>0.05$). Although this is a very modest group, it certainly discards the possibility for these cases to explain HLA genetic findings.

It would be interesting to know if the rs4713462 and rs3828805 loci variants show the same association with HPV positivity in patients without OPSCC that they do in the OPSCC patients. I realize the numbers are too low to do that. Based on their data the authors might make their conclusions clearer if they make a statement about what they would predict in that situation.

To the best of our knowledge, the largest study to date concerning HPV seroprevalence with available genetic data is a population-based study in the UKbiobank that again demonstrated that the seroprevalence of HPV16 E6 and L1 is <1.0% and 4% respectively in individuals without an HPV-related cancer (N=9,503) (Brenner et al.2020, PMID: 33248371). This limits the possibility of assessing the role of HLA variants in individuals without HPV-associated malignancies to only a few hundred individuals, which is clearly underpowered to detect the effects we describe in our analysis. In addition, it is expected that some of HPV-16 E6-seropositive individuals eventually will develop an HPV-driven cancer which makes clearly difficult to rule out this question with sufficient statistical power and confidence.

Using our data, we have reported that oral cavity cancers (OCC), with an estimated HPV prevalence of 5% or less in most populations, are associated with rs3828805 but not with rs4713462. Unfortunately, HPV serology is not available on those samples and we cannot explore these associations as in the same manner as with the OPC data. In our current understanding, rs4713462 seems to be specific of HPV carcinogenesis in the oropharynx and cannot rule out a link to HPV carcinogenesis in other anatomical sites. Since further assumptions would require additional data, we would rather maintain conclusions on these variants within the context of the results we have reported.

The authors' pooling of oral pharyngeal SCC HPV- and oral SCC makes sense and it is gratifying the two groups share similar disease-related genetic associations.

The authors speculation on the possible efficacy of developing a OPSCC treatment with an E6 vaccine is interesting.

We thank the reviewer for these comments.

Line 115. serology is an ideal solution in situation where tumor tissue for direct assessment is not available.

Please remove the word "ideal". Maybe replace with "a good"

We have corrected the wording accordingly.

Reviewer #2, expert in GWAS (Remarks to the Author):

This manuscript aims to determine genetic associations of host genetic polymorphisms with oropharyngeal cancer, through a large case-control GWAS study of HPV+ patients. GWAS significantly identified a signal nearby HLA-B and another signal pointing to the residue 71 on DRB1, in addition to two other signals on chromosome 3 and 4. Interestingly, the associations of HLA class I and II genes associate with the protection against OPC among HPV positive cases, suggesting potentially important protective roles of host immunity mediated through HLA genes. One major concern is with inadequate explorations of multiple genetic variants arising from this highly polymorphic MHC region in which linkage-disequilibrium (LD) is extensive. For example, it is not clear whether or not rs4713462 SNP is high LD with any allele(s) of HLA-B given its proximity, and further it is not clear if the single residue beta71 of DRB1 would be a critical residue contributing to the OPC susceptibility.

It would be important to dig deeper into all genetic variants at nucleotide, residue, and gene allele levels. For such explorations, the conditional analysis is conventionally used, to adjust one risk factor when exploring other factors. However, one should be mindful of its intrinsic limitation, i.e., empirically adjusted factors through a conditional analysis can still have a causal association of importance. Here are specific concerns:

1) Page 4 and Line 110: It is advisable to adopt the official HLA nomenclature for the haplotype DRB1*13:01-DQA1*01:03-DQB1*06:03

We have harmonized the HLA nomenclature across the manuscript accordingly.

2) P6L157: Define MFI

We have defined MFI in the Methods section.

3) P6L169: It seems that two loci on chromosome 3 and 4 are found in the marginal association. It is equally informative to assess these genetic associations, after adjusting for HLA genes.

Thank you for this observation. As requested by the reviewer, we provide HPV(+)-OPC regression model of top associated variants adjusted by the marginal genetic associations in chromosome 3 and 4. As can be seen in Table A, the results are extremely similar to the original results in Tables 2 and Supplementary Table 6, indicating that adjustment by genetic variants on chromosome 3 and 4 makes little difference. There are some minor fluctuations in the p-values and ORs, as expected.

Table A (for response to reviewer only): HPV(+)-OPC top genome-wide significant variants adjusted by marginal associated regions

Model*	Position	Locus	Variant	OR (95%CI)	P value
rs4713462 + 71 -Glu + rs9836282 + rs1516535					
	6p21.33	HLA class I	rs4713462	0.67 (0.59-0.76)	7.1 x 10 ⁻¹¹
	6p21.32	HLA class II	71 -Glu	0.56 (0.46-0.67)	1.7 x 10 ⁻⁹
	3q26.1	609bp 5' of AC092965.1	rs9836282	0.76 (0.68-0.85)	4.2 x 10 ⁻⁷
	4q35.1	ODZ3	rs1516535	1.43 (1.25-1.65)	2.3 x 10 ⁻⁷

*multivariable unconditional logistic regression adjusted for the covariates sex, age and eigenvectors

Alternatively, the reviewer might be referring to possible interactions or trans-regulatory effects given that variants are in different chromosomes and there is no possible linkage disequilibrium between them. Therefore, we have investigated a possible multiplicative genetic interaction by

including HLA genetic variants and either the variant in chromosome 3 or 4 with its corresponding interaction term in the same regression model (multivariable unconditional logistic regression assuming a log-additive genetic or dosage model with age, sex and eigenvectors as covariates). Results in Table B indicate no significant combined effects between top HPV(+)/OPC genome-wide associated variants and those showing marginal associations in the first analysis.

Table B (for response to reviewers only): Analysis of SNP-SNP interactions between HPV(+)/OPC top genome-wide significant and marginal associated regions

SNP 1	POSITION 1	SNP 2	POSITION 2	Interaction* p-value
rs4713462	6p21.33 - HLA class I	rs9836282	3q26.1	0.41
rs4713462	6p21.33 - HLA class I	rs1516535	4q35.1	0.20
71 -Glu	6p21.32 - HLA class II	rs9836282	3q26.1	0.91
71 -Glu	6p21.32 - HLA class II	rs1516535	4q35.1	0.89

* SNP1 and SNP2 indicate the individual SNPs within a given SNP-SNP interaction model. Interaction p-value represents the regression coefficient P values. Adjusted by sex, age and eigenvectors as covariates.

4) P7L185: Given the novel finding of this SNP rs4713462, it seems important to explore its LD relationship with HLA-B and -C, given its proximity. This exploration becomes particularly important, since this SNP is found to be an eQTL with HLA-C.

We thank the reviewer for this comment and have assessed LD relationship between rs4713462 and genetic variants at HLA-B and -C to take these concerns into account. We examined pair-wise LD statistics across SNPs covering in our data HLA-C and B genomic regions in the reference European population from Phase 3 of the 1000 Genomes Project (1000G). In Supplementary Figure 12, pairwise D' and R^2 values were plotted against genomic distance and rs4713462 position highlighted. As it can be seen, SNP rs4713462 showed generally modest correlation (R^2) and coinheritance (D') with SNPs in HLA-B and C coding regions or surrounding areas. If there is any degree of LD, this is more notable with HLA-B. This is in line with our results in Supplementary Figure 13, where rs4713462 is the most significant variant in the region with modest LD and association statistics with the variants covering HLA-C and B. Certainly a limitation to fully address this question is the genetic coverage of the region given that our analyses are based on genotyping data. However, the Oncoarray chip gives a high SNP coverage in the MHC region which helps perform a good quality imputation.

Consistent to the apparent independent assortment of rs4713462 and SNPs in HLA-C region and as discussed in page 7/lines 195-203, our results suggest that HLA-C expression alone is unlikely to be mediating rs4713462 association with HPV(+)/OPC cases. This is based on the fact that other SNPs in the region seem strongly associated with HLA-C expression but not with HPV(+)/OPC risk (Supplementary Figure 3).

The text has been amended in the appropriate sections accordingly with the findings described above (Results page 7, lines 190-195 and Methods page 18, lines 525-527). Supplementary figures 12 and 13 were added to the supplementary material.

5) P7L197-L198: It is not clear how you are able to map multiple nucleotide associations (Supplementary Table 13) to a single residue beta71. Given extensive LD across DRB1-DQA1-DQB1, it is expected that multiple residues may be in high LD with beta 71.

Thank you for pointing this out and apologies for the confusion. We followed several steps to fine-map this region: (1) Independent effects were identified by conditioning on the best associated variants across the MHC region where variation was covered by SNPs, HLA alleles and HLA residues. DRB1 Glu-71 was the best associated variant after conditioning on rs4713462, although others are implicated within each independent LD block as the reviewer noted. (2) We applied haplotype analyses to locate the association boundaries and HLA class II effect was limited to DRB1-DQA1-DQB1. (3) For each gene or region, we looked for the set of significant amino acid positions and we defined the classical alleles with consistent residues at those positions. Therefore, we used a forward selection stepwise regression of significant amino acids, HLA alleles and SNPs within the independently associated region which included the 3 mentioned genes that we now display in Supplementary table 11 and 12. We also included as covariates in these risk models the independent markers previously identified. The aim was to find the best set of amino acids, SNPs and/or HLA alleles that were independently associated with HPV(+)/OPC as judged by the lowest BIC (Bayesian Information Criterion).

Glu-71 accounted for the main part of the effect of DRB1-DQA1-DQB1 as it fitted the data in the conditional model and it has biological significance (supplementary table 12, Model A). However, amino acid (Ala-74) does not seem to be an independent effect on Glu-71 since the effect was not significant ($p > 5 \times 10^{-8}$) and weaker when both are in the model (Supplementary Table 12, model E), whereas the effect was stronger and significant in the absence of Glu-71 (Supplementary Table 12, model C) and in the individual model (Supplementary Table 11). For that reason, we cannot exclude additional loci within the haplotype supporting or contributing additively to risk as probably is happening with the amino acid position 74 in *HLA-DRB1*1301*, also located in the protein binding groove (Supplementary Figure 4).

This point has been now clarified in the corresponding sections (Methods page 17 lines 429-439; Results page 9 lines 234-242; Supplementary tables 11 and 12 were added to the supplementary material).

6) P8L221-222: BIC is commonly used to select variables to build prediction models. In the selection process, the analysis procedure identifies those variables that have overall stronger “empirical associations”, without regarding to underlying biological (causal) associations. Hence, one needs to be cautiously interpreting selected variants, amino acids and haplotypes.

Thank you for this remark. Although observations and functional annotation grant biological plausibility to our findings, we did not do functional assays to pinpoint the association to these variants. We have added a note of caution for our results and have stated the need for future functional analyses of our findings in the main text (line 366-367, page 14).

7) P306L306-307: It is not clear if rs4713462 is accounted for by one or more HLA-B and -C alleles/haplotypes, nor if the single residue beta71 is responsible for HLA-DRB1 association

As discussed in the comment #4 and #5, we included a more detailed explanation of the analysis procedure to select the best combined models (Methods page 17 lines 429-439; Results page 9 lines 234-242). Regarding SNP rs4713462, it is the most relevant variant representing the effect in HLA class I and when other amino acids or HLA alleles were included instead of rs4713462 the models had no support. Therefore, we were not able to identify any HLA-B or -C alleles/haplotypes accounting better for this effect in our data. We have stated this conclusion clearer in the corresponding sections (line 230-231, page 8).

8) P12L312-314: HPV seropositivity is indicative of a humoral responses (humoral immunity), which is somewhat different from the T cell-mediated immunity. However, given identified genetic associations with class I and class II genes, it is plausible both humoral and cell immunities are involved in the protection of OPC. It may be helpful to expand the Discussion.

As additional functional data would be required to discuss further possible connections between humoral and cell-mediated responses with regard to our data, we would rather maintain the current points of discussion.

9) P12L322-324: It would be very helpful to expand discussion on this tantalizing possibility of developing prophylactic vaccines given role of L1 capsid proteins.

More details are now given in the Discussion section (page 13, line 348-358)

10) P16L419-426: It would be informative to carry out “formal haplotype association analysis” of HLA-A, HLA-B/C, HLA-DRB1, HLA-DQA1/B1, with imputed HLA genotypes, with or without other SNPs. If results are informative, they can be included, given their importance. Even if they are secondary, it would be helpful to include them in supplementary tables.

As the reviewer noted, haplotype analyses were only investigated based on HLA alleles. In general, one feature of HLA region makes alleles more efficient to study haplotypes than SNPs. HLA class I and class II alleles are often maintained in the population by balancing selection (e.g. heterozygote advantage) which has resulted in the existence of hundreds of HLA alleles and haplotypes in populations, and the vast majority of which are not common (less than 2% frequency), and collectively account for a significant proportion of the genetic variation. Given the limitations of MHC variation (in terms of density of SNPs and possible low frequency combinations) and our sample size, HLA alleles are likely more adequate to map these evolutionary dynamics than SNPs. Certainly, HLA alleles are themselves multilocus haplotypes, identified by unique combinations of mutations, recombination, and gene conversion with a functional meaning also accounting for variation outside HLA genes.

Due to the above mentioned, we prefer to maintain the focus on HLA alleles to construct haplotypes within this region.

Reviewers' Comments:

Reviewer #1:

Remarks to the Author:

Germline Determinants of Humoral Immune Response To HPV-16 Protect Against Oropharyngeal Cancer by Ferriero-Iglesias et al. identified HPV associated OPC based on HPV E6 seropositivity and other HPV proteins and then link these HPV tumors to differences in HLA sequence and other genome sites. The authors have addressed the issues raised in the initial review.

Minor point - Terminating the immune response - reference 21 is cut short.

Reviewer #2:

Remarks to the Author:

Authors have adequately addressed my comments.

REVIEWERS' COMMENTS

Reviewer #1 (Remarks to the Author):

Germline Determinants of Humoral Immune Response To HPV-16 Protect Against Oropharyngeal Cancer by Ferriero-Iglesias et al. identified HPV associated OPC based on HPV E6 seropositivity and other HPV proteins and then link these HPV tumors to differences in HLA sequence and other genome sites. The authors have addressed the issues raised in the initial review.

Minor point - Terminating the immune response - reference 21 is cut short.

We thank the reviewer for pointing this out. We have now completed reference 21 accordingly.

Reviewer #2 (Remarks to the Author):

Authors have adequately addressed my comments.